# Electroencephalography-Based Brain-Computer Interfaces in Rehabilitation: A Bibliometric Analysis (2013–2023)

**DOI:** 10.3390/s24227125

**Published:** 2024-11-06

**Authors:** Ana Sophia Angulo Medina, Maria Isabel Aguilar Bonilla, Ingrid Daniela Rodríguez Giraldo, John Fernando Montenegro Palacios, Danilo Andrés Cáceres Gutiérrez, Yamil Liscano

**Affiliations:** 1Grupo de Investigación en Salud Integral (GISI), Departamento Facultad de Salud, Universidad Santiago de Cali, Cali 5183000, Colombia; ana.angulo03@usc.edu.co (A.S.A.M.); aguilarbonillamariaisabel@gmail.com (M.I.A.B.); danielamed.internado@gmail.com (I.D.R.G.); 2Specialization in Internal Medicine, Department of Health, Universidad Santiago de Cali, Cali 5183000, Colombia; john.montenegro00@usc.edu.co (J.F.M.P.); danilo.caceres00@usc.edu.co (D.A.C.G.)

**Keywords:** electroencephalography (EEG), Brain-Computer Interface (BCI), rehabilitation, cognitive rehabilitation, motor rehabilitation, neurorehabilitation, bibliometric analysis, EEG-BCI trends

## Abstract

EEG-based Brain-Computer Interfaces (BCIs) have gained significant attention in rehabilitation due to their non-invasive, accessible ability to capture brain activity and restore neurological functions in patients with conditions such as stroke and spinal cord injuries. This study offers a comprehensive bibliometric analysis of global EEG-based BCI research in rehabilitation from 2013 to 2023. It focuses on primary research and review articles addressing technological innovations, effectiveness, and system advancements in clinical rehabilitation. Data were sourced from databases like Web of Science, and bibliometric tools (bibliometrix R) were used to analyze publication trends, geographic distribution, keyword co-occurrences, and collaboration networks. The results reveal a rapid increase in EEG-BCI research, peaking in 2022, with a primary focus on motor and sensory rehabilitation. EEG remains the most commonly used method, with significant contributions from Asia, Europe, and North America. Additionally, there is growing interest in applying BCIs to mental health, as well as integrating artificial intelligence (AI), particularly machine learning, to enhance system accuracy and adaptability. However, challenges remain, such as system inefficiencies and slow learning curves. These could be addressed by incorporating multi-modal approaches and advanced neuroimaging technologies. Further research is needed to validate the applicability of EEG-BCI advancements in both cognitive and motor rehabilitation, especially considering the high global prevalence of cerebrovascular diseases. To advance the field, expanding global participation, particularly in underrepresented regions like Latin America, is essential. Improving system efficiency through multi-modal approaches and AI integration is also critical. Ethical considerations, including data privacy, transparency, and equitable access to BCI technologies, must be prioritized to ensure the inclusive development and use of these technologies across diverse socioeconomic groups.

## 1. Introduction

The use of EEG-based Brain-Computer Interfaces (BCIs) in rehabilitation has significantly expanded over the past decade, particularly in addressing neurological conditions such as stroke, amyotrophic lateral sclerosis (ALS), and spinal cord injuries [1,2,3,4,5]. These conditions affect millions worldwide, with varying incidence and prevalence across regions. For instance, an estimated 15 million people experience strokes globally each year, making it one of the leading causes of long-term disability [6,7]. The rising incidence of these disorders has driven the need for advanced technologies like EEG-based BCIs to offer effective rehabilitation solutions. Recent innovations have focused on enhancing BCI applications for motor, sensory, speech, and cognitive rehabilitation, especially in stroke patients [4]. However, challenges remain, including limited signal throughput, extensive training requirements, and user fatigue, all of which demand further research to improve the clinical effectiveness and usability of these systems [8].

Patients who could benefit from EEG-based BCIs present a wide spectrum of clinical conditions, from complete paralysis to mild motor impairments [9]. By interpreting brain signals and translating them into actionable commands, EEG-based BCIs enable these patients to interact with their environment in unprecedented ways [10,11,12]. This includes controlling prosthetics, communicating through assistive devices, and performing daily tasks, all aimed at improving their quality of life and autonomy [1,13].

Despite their potential, the implementation of EEG-based BCIs in rehabilitation faces several challenges, such as variability in brain signals, the need for personalized calibration, and maintaining a stable and reliable connection. Additionally, there is ongoing debate within the scientific community regarding the long-term efficacy of BCIs and their real impact on rehabilitation, emphasizing the need for more rigorous and comprehensive research in this field [14,15].

Currently, BCI-based treatment approaches are in an experimental phase, with several clinical trials underway to evaluate their efficacy and safety [14]. While these studies have shown promising results, they also underscore the necessity for standardized management protocols and a deeper understanding of the mechanisms behind effective EEG-based BCI use. Integrating BCIs into rehabilitation programs will require interdisciplinary collaboration and personalized approaches for each patient [16,17].

This study aims to analyze the development and trends in global research on BCIs in rehabilitation from 2013 to 2023, using bibliometric tools. The research seeks to provide a comprehensive overview of the current state of the field, identify key areas of progress, and highlight knowledge gaps requiring further exploration. Through this analysis, we aim to enhance understanding of how BCIs can be more effectively integrated into rehabilitation programs, emphasizing the necessity for continued research to optimize their application and maximize benefits for patients.

## 2. Materials and Methods

### 2.1. Design and Data Collection

This study conducted a bibliometric analysis to evaluate the development and trends in global research on BCIs in the context of rehabilitation from 2013 to 2023. Data were collected from recognized academic databases including Web of Science (WoS), PubMed, Science Direct, and Scopus. The search strategy employed a comprehensive combination of keywords related to BCIs and rehabilitation.

### 2.2. Search Strategy

The search was restricted to English-language publications to maintain consistency and relevance. The search terms included combinations such as:

(“Electroencephalography” OR “EEG” OR “Electroencephalogram”) AND (“Brain-Computer Interface” OR “BCI” OR “Neural Prostheses”) AND (“Rehabilitation” OR “Motor Rehabilitation” OR “Cognitive Rehabilitation” OR “Sensory Rehabilitation” OR “Speech Rehabilitation” OR “Neurorehabilitation”) AND (“Stroke” OR “Amyotrophic Lateral Sclerosis” OR “ALS” OR “Spinal Cord Injuries” OR “Neurological Disorders”).

### 2.3. Selection Criteria

#### 2.3.1. Inclusion Criteria

Studies focusing on the application of EEG-based BCIs in rehabilitation, encompassing all forms and uses of EEG-BCI technology within this context.Publications in peer-reviewed journals indexed in databases such as Web of Science (WoS), PubMed, Science Direct, and Springer.Inclusion of both primary research articles and review papers that provide significant insights into the implementation, effectiveness, technological innovations, and theoretical advances in EEG-BCI-based rehabilitation.Studies published between 2013 and April 2023, ensuring coverage of the most recent trends and developments.Articles published in English.

#### 2.3.2. Exclusion Criteria

Documents such as editorials, commentaries, brief notes, press releases, or book chapters.Publications not indexed in the specified databases, indicating potentially lower relevance or impact.

### 2.4. Data Establishment and Processing

The extracted data were imported into Microsoft Excel version 2019 (Microsoft, Redmond, WA, USA) for further processing. Duplicate records were removed, and two researchers (I.D.R.G. and A.S.A.M.) independently screened the remaining studies based on predefined inclusion and exclusion criteria focused on EEG-based BCIs. Discrepancies in selection were resolved by a third evaluator (Y.L.). Key data points such as Year of publication, Manuscript title, Country of origin, Journal name, Quartile ranking (Q1–Q4), Citation count, Rehabilitation category, Most frequently used EEG-based BCI method, Average duration of treatment (weeks), and Reported limitations and health risks were recorded for analysis.

### 2.5. Analytical Tools and Procedures

Microsoft Excel version 2019 (Microsoft, Redmond, WA, USA) and R (ggplot2 and Bibliometrix packages, version 4.3.1) were used for data management and preliminary analysis, including generating descriptive statistics and visualizations.Bibliometrix was used to analyze cooperation networks, keyword co-occurrences, and thematic mapping specific to the research landscape of EEG-based BCIs in rehabilitation.

### 2.6. Data Analysis

The study employed bibliometric techniques to analyze publication output trends, citation patterns, and collaboration networks. The analysis included:Temporal trends in publication and citation counts.Geographic distribution of research outputs by continent and country.Keyword co-occurrence analysis to identify emerging research themes.Evaluation of research output by leading authors and institutions.

## 3. Results

The following sections will present a comprehensive overview of the key aspects of our bibliometric analysis on EEG-based BCIs in rehabilitation. First, we will introduce the methodology using a flowchart to clarify the selection process of the studies included in this review. This is followed by an examination of annual publication trends, providing insight into the field’s growth over time. Next, we will present the geographic distribution and collaboration networks to understand the global expansion and the key contributors to EEG-BCI research. Subsequently, we will highlight the most influential publications and leading institutions driving advancements in this area. Following this, we will describe the different study designs and methods used in rehabilitation to contextualize the application of EEG-BCIs. Finally, we will conclude with a keyword analysis and thematic map to identify emerging research themes and future directions in the field.

### 3.1. Search Results and Study Selection

The initial search yielded a total of 2640 records from four databases: PubMed (375), WoS (488), Science Direct (938), and Scopus (839) (see Figure 1). After removing 1169 duplicate records, 1471 records were screened. Of these, 938 records were excluded due to being editorials, books, non-English articles, articles outside the 2013–2023 range, conference papers, protocols, or textbooks. Finally, 533 studies were included in the bibliometric analysis.

### 3.2. Annual Trends in EEG-BCI Research

Figure 2 provides a detailed analysis of document production and citations over the period from 2013 to 2023. In Figure 2A, the number of articles published per year shows a clear upward trend, with noticeable growth beginning around 2016 and a peak in publications observed in 2022. This increasing trend in research output reflects the growing interest and investment in this scientific field. In panel Figure 2B, the average number of citations per year is highlighted, showing some fluctuations but generally following a declining trend after an initial peak. This suggests that while the number of publications has grown, the average citations per paper have not kept pace, possibly due to the typical lag in citation accumulation.

Figure 2C illustrates the distribution of document types, revealing that 85.7% of the documents are research articles, whereas 14.3% are reviews, indicating a predominant focus on primary research contributions in the field, with a smaller proportion dedicated to the synthesis of existing knowledge.

Finally, Figure 2D presents the number of documents published per journal, with the *IEEE Transactions on Neural Systems and Rehabilitation* and *Frontiers in Neuroscience* being the most prolific sources. These results collectively show that the field has experienced substantial growth in research output, although citation impact may vary across years and types of publications.

Continuing with the results related to journals, Figure 3 illustrates the cumulative occurrences of publications in the top 5 journal sources from 2013 to 2023. The graph shows a clear upward trend in the number of publications across all sources, with *Frontiers in Human Neuroscience* and *IEEE Transactions on Neural Systems and Rehabilitation Engineering* leading in cumulative occurrences. These two journals show a significant increase in the number of publications, particularly after 2016. The remaining sources, *Frontiers in Neuroscience*, *Journal of Neural Engineering*, and *Journal of NeuroEngineering and Rehabilitation*, also demonstrate steady growth, albeit at a slower pace compared to the top journals.

### 3.3. Geographic Distribution of EEG-BCI Research

In the section on the geographic distribution of EEG-BCI publications, most research originates from China and the United States, as shown in Figure 4. Figure 4A highlights regions with the highest scientific output, with darker blue shades indicating more publications. Figure 4B emphasizes international collaborations, showing strong connections between leading countries like the U.S., China, and the European nations. Figure 4C ranks countries by the number of published documents, distinguishing between Single-Country Publications (SCP) and Multiple-Country Publications (MCP). China leads in total publications, followed by the United States and Germany, indicating significant international cooperation in the EEG-BCI field.

In relation to Latin American countries researching EEG-BCI, only Brazil and Mexico are present in the top 20 of scientific production in the EEG-BCI field, as shown in Figure 4. In Figure 4C, Brazil stands out with a moderate number of publications, indicating both Single-Country Publications (SCP) and Multiple-Country Publications (MCP), reflecting its participation in international collaborations. Mexico also appears on the map, though with a smaller contribution. These trends suggest that while EEG-BCI research is emerging in Latin America, its impact remains considerably lower compared to major scientific powers like China and the United States. The presence of Brazil and Mexico can be attributed to their growing investment in neuroscience and rehabilitation technologies, often in collaboration with other countries, indicating a potential for future development as international cooperation strengthens [18,19].

Figure 5 illustrates the annual growth of EEG-BCI-related publications for the top five contributing countries from 2013 to 2023. China and the United States show a notable increase in output, with China experiencing exponential growth, reaching over 350 articles by 2023. Japan, while behind China, also demonstrates a sharp rise in recent years. In contrast, Germany, Italy, and Japan show more gradual increases in their publication output, maintaining consistent but lower numbers compared to China and the U.S.

This trend highlights China’s leading role in the field, likely driven by increased governmental and institutional support for neurotechnology research. The rise in U.S. publications can be attributed to a long-standing focus on innovation in biomedical engineering and neuroscience [20,21]. The slower yet steady growth in European countries reflects their sustained investment in EEG-BCI technologies, though on a smaller scale compared to the leading nations [20].

### 3.4. Key Publications and Journals in EEG-BCI

In this section, we summarize the most highly cited publications along with their key metrics, as outlined in Table 1. These studies represent foundational contributions to the field of EEG-based BCIs in rehabilitation, offering critical insights into the advancement and application of these technologies.

The data reveals that the most influential studies come primarily from institutions in the USA, Germany, Switzerland, and Singapore. These publications consist largely of RCTs and review articles, emphasizing the field’s focus on rigorous evidence-based research and thorough literature analysis. Most of these studies appear in top-tier, Q1-ranked journals such as *Annals of Neurology*, *Nature Reviews Neurology*, and *Nature Communications*, underscoring the high standards of research quality and scientific rigor in EEG-BCI studies.

Prominent institutions like Duke University, the University of Tübingen, and Nanyang Technological University lead research efforts in this domain, fostering international collaborations that integrate expertise across neuroscience, robotics, and rehabilitation engineering. For instance, the work by Biasiucci et al. 2018 [22] at École Polytechnique Fédérale de Lausanne exemplifies how combining engineering with clinical research can drive significant advances in post-stroke rehabilitation.

Additionally, many of these studies benefit from substantial funding from both governmental and private sources, highlighting the significant investment in EEG-BCI research. Prestigious grants from organizations like the NIH, A*STAR, and other national research councils have enabled groundbreaking research and facilitated international collaboration. This financial backing reflects global recognition of the potential for EEG-BCI technologies to revolutionize rehabilitation practices.

Funding plays a pivotal role in determining the scope, quality, and impact of research. Adequate funding allows researchers to conduct more comprehensive studies, access cutting-edge technologies, and collaborate across disciplines [23]. In EEG-BCI research, large-scale funding often supports interdisciplinary efforts, enabling breakthroughs that blend neuroscience, engineering, and clinical practice. Moreover, well-funded research tends to be published in prestigious journals, enhancing its visibility and influence [15]. The availability of resources directly affects the ability to address complex questions, produce reliable results, and contribute meaningfully to the scientific community [23].

**Table 1 sensors-24-07125-t001:** Summary of Top 10 Highly Cited Studies on EEG-Based BCIs in Rehabilitation.

Authors and Year of Publication	Manuscript	Country	Affiliation	Funding	Type of Study	Journal	Quartil	Citations
Ramos-Murguialday et al., 2013 [24]	Brain-Machine Interface-Based Gait Protocol for Paraplegic Patients	USA	Duke University	Not specified	Review	*Ann. Neurol.*	Q1	739
Chaudhary et al., 2015 [1]	Brain–Computer Interfaces for Communication and Rehabilitation	Germany, Switzerland, Spain	University of Tübingen, Wyss-Center for Bio- and Neuro-Engineering, TECNALIA Health Department	Not specified	Review	*Nature Reviews Neurology*	Q1	583
Abiri et al., 2018 [25]	A Comprehensive Review of EEG-based Brain-Computer Interface Paradigms	USA	University of California, San Francisco/Berkeley; University of Tennessee	N/A	Review	*Journal of Neural Engineering*	Q2	545
Pichiorri et al., 2015 [26]	Brain-Computer Interface Boosts Motor Imagery Practice During Stroke Recovery	Italy	IRCCS Fondazione Santa Lucia, Sapienza University of Rome, Italy	Not specified	Randomized Controlled Trial (RCT)	*Annals of Neurology*	Q1	448
Ang et al., 2015 [9]	A Randomized Controlled Trial of EEG-Based Motor Imagery BCI Robotic Rehabilitation for Stroke	Singapore	Institute for Infocomm Research, A*STAR, Tan Tock Seng Hospital, Singapore	Enterprise Challenge Grant, Prime Minister’s Office, Singapore; Science and Engineering Research Council, A*STAR	Randomized Controlled Trial (RCT)	*Clinical EEG and Neuroscience*	Q2	385
Lebedev and Nicolelis, 2017 [27]	Brain-Machine Interfaces: From Basic Science to Neuroprostheses and Neurorehabilitation	USA	Duke University	Not specified in provided text	Review	*Physiological Reviews*	Q1	359
Biasiucci A. et al., 2018 [22]	Brain-Actuated Functional Electrical Stimulation Elicits Lasting Arm Motor Recovery After Stroke	Switzerland	École Polytechnique Fédérale de Lausanne, University of Geneva	SUVACare, Clinique Romande	Randomized Controlled Trial (RCT)	*Nature Communications*	Q1	349
Donati et al., 2016 [28]	Long-Term Training with a Brain-Machine Interface-Based Gait Protocol Induces Partial Neurological Recovery in Paraplegic Patients	Brazil, USA, Switzerland	Neurorehabilitation Laboratory, Duke University	NIH, AACD, etc.	Case series	*Scientific Reports*	Q1	313
Frolov et al., 2017 [29]	Post-Stroke Rehabilitation Training with a Motor-Imagery-Based Brain-Computer Interface (BCI)-Controlled Hand Exoskeleton: A Randomized Controlled Multicenter Trial	Russia	Pirogov Russian National Research Medical University	Russian Ministry of Education and Science	Randomized Controlled Trial (RCT)	*Frontiers in Neuroscience*	Q2	262
Ang et al., 2014 [30]	Brain-Computer Interface-Based Robotic End Effector System for Wrist and Hand Rehabilitation	Singapore	Nanyang Technological University	No conflicts of interest reported, funded by the National Medical Research Council	Randomized Controlled Trial (RCT)	*Frontiers in Neuroengineering*	Q2	254

### 3.5. Institutional Contributions to EEG-BCI Research

The analysis of institutional contributions to EEG-BCI research reveals the significant roles played by key institutions in advancing the field. As detailed in Table 2, the University of Wisconsin–Madison leads with 36 published articles, followed closely by Aalborg University and the University of Tübingen, each contributing 30 publications. Aalborg University’s impact is particularly noteworthy, as it has become a major player in developing innovative BCI applications, especially for rehabilitation [31]. The university’s strong interdisciplinary research environment, which integrates engineering, neuroscience, and health sciences, enables it to bridge theoretical advancements with practical, clinical applications. This focus positions Aalborg as a central contributor to improving treatments for patients with neurological conditions. Other notable contributors include the University of California and Fudan University, highlighting the international scope of EEG-BCI research, with significant input from institutions in North America, Europe, and Asia.

Figure 6 highlights the publication trends by affiliation from 2013 to 2023, showing steady growth in output from these leading institutions. The University of Wisconsin–Madison and Aalborg University have demonstrated particularly strong growth in recent years, signaling an increasing focus on and investment in EEG-BCI research. This surge is in line with the global rise of interest in neurorehabilitation technologies and BCIs, underscoring the strategic importance of these areas [15].

Figure 7 shifts focus to individual author contributions and collaboration networks. Figure 7A identifies leading contributors such as Niels Birbaumer and Kai Keng Ang, who have published the highest number of documents in the field. Figure 7B illustrates the collaboration networks, showcasing clusters of authors working together on EEG-BCI projects. Birbaumer and Ang are central figures in these networks, reflecting the interdisciplinary and international nature of research in this domain. Figure 7C tracks the productivity of key authors over time, showing consistent contributions from figures like Birbaumer, Ang, and Wang X., further cementing their influence in EEG-BCI research.

Niels Birbaumer, a renowned neuroscientist, has made groundbreaking contributions to the field of BCI. His research primarily focuses on neuroplasticity, learning, and the development of non-invasive BCIs to assist patients with severe neurological conditions, such as locked-in syndrome. Birbaumer’s work has been instrumental in developing EEG-based BCIs that decode neural signals in patients who are completely paralyzed but cognitively aware [1,12,24]. His collaboration with the Wyss-Center for Bio- and Neuro-Engineering in Geneva, Switzerland, further extends his impact. The Wyss-Center is a leading institution dedicated to neurotechnology and neuroscience research, developing innovative tools and therapies for people with neurological disorders. Its focus on projects like neural implants and BCIs aligns with Birbaumer’s vision of using neurotechnology to restore communication and improve the quality of life for individuals with severe neurological impairments [32].

Similarly, Kai Keng Ang has significantly advanced the field of BCIs, with a particular focus on motor imagery-based BCIs for stroke rehabilitation. Ang has been instrumental in developing non-invasive BCI systems that assist stroke survivors in regaining motor function. His work integrates BCIs with robotic systems, allowing brain signals captured via EEG to control robotic devices or virtual environments, enhancing rehabilitation outcomes through neurofeedback and repetitive task training. Ang’s research demonstrates the potential of EEG-based BCIs for real-world clinical applications, helping patients recover motor skills through brain-driven therapy [9,30,33]. Affiliated with the Institute for Infocomm Research (I^2^R) under A*STAR in Singapore, Ang has led numerous studies that combine BCI technology with artificial intelligence (AI) and robotics [34]. His work focuses on validating BCI systems in clinical settings, often through randomized controlled trials (RCTs), which are crucial for demonstrating the efficacy of these systems in stroke rehabilitation. His contributions have significantly influenced the application of BCIs in neurorehabilitation, advancing both the understanding and practical use of these technologies in therapeutic settings [33].

### 3.6. Study Designs and BCI Methods in Rehabilitation

In the section Study Designs and BCI Methods in Rehabilitation, Figure 8 provides a comprehensive overview of the various methodologies and research fields contributing to EEG-based BCI research in rehabilitation.

Figure 8A expands on the information presented in Figure 2C, emphasizing the distribution of study designs used in the analyzed publications. Among the most frequent study types are RCTs and systematic reviews. Systematic reviews are crucial for synthesizing existing knowledge and identifying gaps in the literature [35]. However, the relatively lower number of RCTs is notable. RCTs are considered the gold standard for validating new interventions and technologies, and their increase in this field would provide more robust evidence of the efficacy and safety of BCI applications, especially in clinical settings. These trials are essential for moving BCIs toward broader clinical acceptance, particularly in rehabilitation, where clear, evidence-based outcomes are critical for integrating these technologies into routine therapeutic practices [36,37].

Figure 8B demonstrates the Proportional Distribution of Academic Fields contributing to BCI research. Neuroscience (23.4%) and Medicine (23.3%) are the leading contributors, reflecting the interdisciplinary nature of BCI development. Engineering (18.9%) also plays a significant role, highlighting the importance of technological advancements in BCI applications. The involvement of various fields such as computer science, psychology, and health professions emphasizes the collaborative efforts needed to drive BCI research forward. This interdisciplinary approach ensures that both the technical and clinical aspects of BCI are considered, leading to more robust and practical solutions in rehabilitation [38,39].

Figure 8C illustrates the Frequency of Different Rehabilitation Types addressed in the research, with motor rehabilitation being the most frequently studied area. This focus is crucial as motor rehabilitation aims to improve the quality of life for individuals suffering from neurological impairments such as stroke or spinal cord injuries. BCIs, particularly in motor rehabilitation, have shown promising results, allowing patients to engage in neurofeedback and task-oriented training, which supports the recovery of motor control. This concentration on motor rehabilitation reflects the field’s current emphasis on helping patients regain essential movements, which is a primary goal in many rehabilitation programs [40]. While most research has focused on stroke rehabilitation, there is potential for BCI applications in other neurological conditions, such as spinal cord injury [41,42].

However, the relatively fewer studies on cognitive, sensory, and speech rehabilitation highlight areas for future research. Expanding the number of RCTs in these underexplored domains could broaden the applicability of BCI technologies. This would ensure that individuals with various neurological conditions benefit from the advancements in BCI research, just as those in motor rehabilitation have [43]. Cognitive Multisensory Rehabilitation has demonstrated improvements in upper limb function and sensorimotor recovery, with changes in brain connectivity observed post-intervention (Winckel et al., 2020) [44]. Challenges in study design and analysis for cognitive rehabilitation research have been identified, with recommendations for multiperiod, multiphase crossover designs [45].

### 3.7. Keyword Analysis and Research Themes

Keyword analysis is essential for identifying evolving trends and priorities in EEG-BCI research, offering insights into key focus areas and emerging topics. This helps researchers align their work with current advancements and funding opportunities, while also revealing gaps in the literature. In EEG-BCI, keyword trends reflect the field’s multidisciplinary nature, combining neuroscience, rehabilitation, and technology development. The growing prominence of rehabilitation-related terms highlights the increasing application of BCI technologies in clinical settings, especially for patients recovering from neurological conditions like strokes [46,47].

Figure 9 illustrates the most commonly used keywords and their trends in EEG-BCI publications. Figure 9A presents a word cloud where the size of the words represents their frequency in the analyzed publications. Unsurprisingly, terms like “electroencephalography” and “brain-computer interface” are dominant, reflecting the core focus of the research. Other significant terms include “stroke rehabilitation”, “human”, and “cerebrovascular accident”, which highlight the clinical applications of EEG-BCI technologies, particularly in rehabilitation for stroke patients. Figure 9B shows the cumulative occurrences of selected keywords from 2013 to 2023. “Electroencephalography” and “brain-computer interface” exhibit steady growth, reaffirming their centrality in this field. “Stroke rehabilitation” and “cerebrovascular accident” also show increasing trends, which aligns with the growing interest in using BCI technologies for rehabilitation purposes. The consistent rise in keywords related to gender, such as “female” and “male”, points to the growing attention on gender differences in EEG-BCI research, possibly addressing different rehabilitation outcomes or methodologies for males and females.

Figure 9C provides a timeline of emerging keywords, revealing when specific terms gained prominence in the literature. Terms like “motor imagery”, “fugl-meyer assessment for upper extremity”, and “stroke rehabilitation” are particularly important, as they reflect the shift towards more specialized applications of EEG-BCI systems [15,48]. The emergence of newer terms in recent years indicates that the field is evolving to explore innovative applications and more precise methodologies, particularly in motor rehabilitation and neuroplasticity. The field has evolved from early proof-of-principle demonstrations to exploring innovative applications, with a growing emphasis on specialized methodologies in motor rehabilitation and neuroplasticity [46]. Machine learning and deep learning algorithms have been increasingly applied to enhance BCI-driven rehabilitation assessment and outcomes [49].

### 3.8. Thematic Analysis and Future Research Directions

Thematic analysis and future research directions play a critical role in bibliometric studies of EEG-BCI research. This type of analysis helps identify the key themes driving current research and highlights areas that may require further exploration. By examining keyword clusters and their relationships, researchers can identify emerging trends, understand the evolution of certain research topics, and guide their own studies to align with or fill gaps in the existing literature.

Figure 10 provides a detailed thematic and factorial analysis of BCI research keywords. Figure 10A presents a thematic map, revealing the development and relevance of different research themes in the field. It visually demonstrates how central topics, such as “brain-computer interface” and “human”, are interconnected with more specialized areas like “cerebrovascular accident” and “rehabilitation.” This clustering highlights both well-established and emerging themes in BCI research.

Figure 10B offers a factorial analysis plot, displaying the relationships between key terms and their roles in shaping the research landscape. The connections among terms like “movement physiology”, “robotics”, and “neurorehabilitation” emphasize the interdisciplinary nature of BCI research, particularly in rehabilitation. This analysis not only identifies the most influential research areas but also suggests potential future directions, such as refining the application of BCI technologies in clinical rehabilitation for neurological conditions like stroke.

For Figure 10C, the graph depicts the spatial arrangement of keywords, indicating their relative importance and connections in the EEG-BCI field. Terms like “clinical article”, “electroencephalogram”, and “stroke” are positioned prominently, suggesting their central role in recent studies. The clustering of terms related to “movement”, “imagery”, and “brain-computer interface” emphasizes the critical focus on motor rehabilitation. The dimensional analysis provides insights into how different research themes intersect and contribute to the broader scope of BCI research, revealing potential areas for further investigation and refinement.

Building on these findings, emerging themes, such as gender differences and specific neurophysiological responses, offer valuable insight into underexplored areas that could shift research priorities. Addressing these gaps fosters new studies, encourages interdisciplinary approaches, and deepens the understanding of how BCIs can be integrated into diverse therapeutic applications [39,50].

Although the thematic analysis did not explicitly highlight AI and ethics, these areas are expected to play a crucial role in the future of EEG-BCI research. AI and machine learning are set to play an essential role in decoding brain signals and predicting user intentions, making BCIs more intuitive and responsive. By leveraging AI, EEG-based BCIs can become more adaptive, improving their ability to tailor neurorehabilitation programs to individual needs, and optimize rehabilitation outcomes in real time. This evolution will enhance the system’s ability to process vast amounts of data, refine signal accuracy, and create more personalized, effective interventions [51,52].

## 4. Discussion

The purpose of this study is to provide a comprehensive overview of the current state of research on EEG-based BCIs in rehabilitation over the past decade, utilizing bibliometric tools. Through this analysis, the study aims not only to identify patterns of scientific output and the key players in the field but also to highlight future opportunities and remaining challenges in the application of these technologies.

The findings reveal an exponential growth in scientific publications since 2016, underscoring the increasing significance of EEG-BCIs in clinical contexts, particularly in neuromotor, sensory, and cognitive rehabilitation. In the early 2010s, research on EEG-based BCIs primarily focused on demonstrating the feasibility of these technologies in motor rehabilitation [53]. This initial focus was crucial for validating the effectiveness of brain-computer interfaces, particularly in restoring motor functions in patients with severe neurological injuries, such as strokes [54]. During this time, BCI technologies began to show their potential in translating brain signals into physical actions, using neurofeedback systems and robotics to enhance patient mobility. This period of development laid the groundwork for more advanced research that emerged after 2016, when more complex and applied studies began to be published [55].

Additionally, the analysis highlights contributions from various regions around the world, with a concentration of publications in specific geographic areas, and shows a predominance of experimental studies over reviews, emphasizing the priority placed on generating new knowledge in this field. Geographical patterns in scientific production were identified, with a higher concentration of publications in Asia, Europe, and North America, with China, the United States, and European countries like the UK, Germany, and Italy being the main contributors. This increase reflects greater attention and resources dedicated to this field, driven in part by technological advancements and the growing acceptance of BCIs in clinical applications [20]. Moreover, the interest in these technologies has fostered interdisciplinary collaboration among engineers, medical professionals, and behavioral scientists, who work together to improve the effectiveness and functionality of BCIs. This synergy has been fundamental in advancing the integration of BCIs into clinical settings, promising significant improvements in the quality of life for patients with neurological disabilities [56,57].

These results were also observed in the bibliometric analysis by Yin et al., 2022 [58]. In contrast, South American countries showed significantly lower participation in BCI research compared to other regions. This disparity may be due to several factors, including a lack of advanced technological infrastructure, lower research funding, and limitations in training specialists in critical interdisciplinary areas for BCI development [20,59]. Moreover, international cooperation and collaboration networks with leading countries in the field are less frequent, limiting access to cutting-edge technologies and the opportunity to develop joint projects. To enhance the visibility and impact of South American countries in BCI research, it is essential to promote policies that invest in science and technology, facilitating access to advanced equipment and the training of specialized researchers [60]. While Brazil has shown moderate growth in its participation, contributions from other parts of the region remain limited. To advance the development of EEG-BCIs in Latin America, it is essential to increase investment in science and technology, encourage international collaboration, and implement policies that support the training of researchers specialized in neurotechnology. The main factors hindering scientific progress include insufficient investment in science and technology, inadequate research infrastructure, limited access to grants, and economic instability [61,62]. Building collaborative networks with leading institutions, such as universities in China and the United States, would facilitate access to cutting-edge technologies and enhance the impact of research in this region [18,20].

Highly cited articles, such as Ramos-Murguialday et al., 2013 [63], in the study highlighted the efficacy of brain-machine interfaces in improving motor function in patients with severe paresis through the contingent control of hand and arm movements via desynchronization of sensorimotor rhythms. This research underscores the clinical relevance of BMIs, particularly in inducing functional improvements when paired with physiotherapy. The impact of these findings, reflected in the improvement of Fugl-Meyer scores and associated cortical reorganization, represents a transformative step in neurorehabilitation for chronic stroke patients without residual movement capacity.

In this analysis, prolific authors such as Ang K.K., Birbaumer N., and Jochumsen M. are identified as key contributors to the field, with Ang K.K. leading in the number of documents published. Yin et al., in a 2022 study [58], noted the central role of Birbaumer N. and Ang K.K. in connecting different research groups and fostering collaboration across multiple teams. This highlights their pivotal contribution to advancing research through extensive international partnerships. From 2011 to 2021, influential authors like Birbaumer N., Ang K.K., and Jochumsen M. have consistently contributed to the field, with notable peaks in collaborative efforts.

Various BCI methods have been employed in studies, with EEG being the most utilized technique. However, other techniques such as fMRI, NIRS, ECoG, and LFP have also been implemented, highlighting the predominance of EEG due to its wide application and accessibility in research [48,64,65,66,67]. Barboza et al. 2024 [68] supports the consideration of EEG as a primary diagnostic tool for clinical entities associated with brain physiology, emphasizing its low cost, easy accessibility, and reduced risk, making it indispensable in recent clinical interventions. Other studies, such as Jeste et al. 2015 [69], suggest that interventions focused on social engagement can improve children’s attention and interest in social information, which can be quantified by EEG oscillatory patterns, providing important clinical value.

Additionally, the proportional distribution of rehabilitation types indicates that motor rehabilitation is the primary focus of research, followed by sensory rehabilitation and cognitive rehabilitation. This highlights the predominant emphasis on these areas within BCI research, which holds significant clinical applications. This emphasis on motor rehabilitation may be related to the availability of established technologies and protocols, such as the use of EEG and neurofeedback, which have demonstrated efficacy in improving motor functions [48]. While motor rehabilitation receives substantial attention, sensory rehabilitation is gaining prominence due to the increasing recognition of its role in enhancing the quality of life for patients with sensory impairments, such as vision or hearing loss [70], or those with somatosensory deficits after stroke. Sensory rehabilitation within BCI research involves the development of interfaces that can restore or augment sensory perception by creating artificial feedback loops between the brain and external stimuli. For instance, BCIs have been used to deliver sensory feedback through electrical or tactile stimulation, enabling individuals to regain a sense of touch, temperature, or proprioception, often in tandem with motor rehabilitation efforts. The focus on sensory rehabilitation is also driven by advancements in neuroplasticity studies, which show that the brain can reorganize and adapt to new forms of sensory input [71,72].

On the other hand, cognitive rehabilitation, which often requires more personalized and complex approaches, could benefit from increased research and development to optimize its applications [44,73]. Furthermore, BCIs are increasingly being explored for mental health applications, including neurofeedback techniques that help individuals manage conditions like anxiety, depression, epilepsy, and ADHD. By modulating brain activity, these interventions can potentially improve emotional regulation and cognitive functions, offering new avenues for treatment [74,75]. Cognitive BCI research has seen progress in developing interfaces that support attention enhancement and memory recall, particularly useful for individuals with cognitive impairments due to conditions like stroke or traumatic brain injury [76,77,78]. These advancements are pushing the boundaries of what BCIs can achieve in clinical settings, making them a viable option for a broader range of therapeutic applications [79].

Yin et al. 2022 [58] highlight the bidirectional capability of BCIs, emphasizing their excellent potential for neurorehabilitation, and promoting the advancement in the development and application of deep learning algorithms and neural networks in this field. Bensmaia & Miller 2014 [80] identify key engineering challenges for the next generation of BCIs, such as incorporating somatosensory feedback and direct limb impedance control, suggesting that the clinical acceptance of these technologies will depend on the success of these developments over the next 5 to 10 years. Other authors, like Stoeckel et al. 2014 [81], note that the use of neurofeedback with real-time fMRI (rt-fMRI) can aid in developing safe, effective, and personalized therapies for brain disorders like pain, addiction, phobia, anxiety, and depression, clarifying fundamental brain-behavior relationships for understanding and treating these disorders [81].

Romagosa et al. 2020 [82] analyzes the effects of BCI therapy on upper limb motor rehabilitation in stroke survivors, highlighting significant motor function improvements that persist after rehabilitation. This suggests that each patient should be profiled using a comprehensive battery of clinical and neurophysiological assessments to design a personalized rehabilitation program based on their functional needs and neurological biomarkers. Neurofeedback plays a crucial role in BCI-based rehabilitation, as indicated by Bruner et al. 2024 [83] in their study on BCI training that combines motor imagery with functional electrical stimulation (FES).

However, the widespread use of BCIs for neurorehabilitation still faces challenges, such as system inefficiencies and learning speed, which can be improved with multimodal or multi-stage approaches that use more sensitive neuroimaging technologies [14]. The measurement of additional neuronal systems and performance in non-motor tasks is essential to demonstrate the specificity or transferability of improvements in cognitive and motor domains [84]. BCI research, primarily in China, the United States, and Italy, has focused on rehabilitation, avoiding damaged neural pathways. Given the high prevalence of cerebrovascular diseases, which are the leading cause of disability, most reviews focus on advances in neuroscience and algorithms to improve the stability, precision, and speed of BCI systems, with growing social acceptance [47].

In this context, the analysis of keyword trends in EEG-BCI research provides significant insights into emerging areas and future directions. Terms such as “stroke rehabilitation”, “motor imagery”, and “fugl-meyer assessment for upper extremity” have seen increasing use in recent years, indicating new research focuses and methodologies being explored [85]. The growing prominence of terms related to rehabilitation underscores a trend toward applying EEG-BCI systems in clinical settings, particularly for motor and sensory recovery. This expansion highlights the increasing integration of advanced machine learning algorithms, such as deep learning, to improve the accuracy and effectiveness of BCI-driven rehabilitation [49,86]. Furthermore, the rise of gender-related terms like “female” and “male” reflects an emerging interest in tailoring BCI applications to account for gender-specific differences in treatment outcomes. This trend signals a shift toward a more nuanced and individualized understanding of brain-computer interactions, potentially leading to more precise and personalized therapeutic applications, both in clinical and non-clinical contexts [87].

The prominence of keywords like “human”, “brain computer interface”, and “electroencephalography” underscores the continued emphasis on human-centered studies and the development of BCI technologies that facilitate interaction with external devices through brain signals. The thematic and factorial analysis of keywords highlighted key research areas, such as motor rehabilitation and the use of EEG as a predominant method. This analysis also reveals how established themes like “rehabilitation” and “cerebrovascular accident” are interconnected with emerging areas such as robotics and movement physiology, emphasizing the interdisciplinary nature of BCI research [49,88]. Additionally, the clustering of keywords into categories like ‘clinical article’, ‘brain computer interface’, and ‘human’ indicates a structured research landscape where clinical applications, technical advancements, and human-centered studies are central themes. This ongoing evolution and diversification in research focus are crucial for enhancing the visibility and impact of BCI technologies.

Although AI and ethics were not explicitly highlighted in the thematic analysis, they are expected to play a pivotal role in the future of EEG-BCI research. AI, particularly through machine learning, is set to significantly enhance the decoding of brain signals and the prediction of user intentions, making BCIs more intuitive and adaptive. By integrating AI, EEG-BCIs will provide more personalized neurorehabilitation programs, optimizing outcomes in real time and improving signal accuracy. This advancement will refine the system’s capacity to process large data sets and enable more effective, tailored interventions [51,89,90].

These technologies will improve the precision of brain signal interpretation, allowing for the customization of neurorehabilitation therapies and real-time adjustments based on the patient’s needs. AI has the potential to reduce noise in EEG data, making these systems more accessible and effective for clinical use. Additionally, the use of big data will enable the long-term analysis of brain activity patterns, offering clinicians more robust tools for monitoring and making therapeutic decisions [91,92].

Beyond technological advancements, ethical considerations are paramount. As EEG-BCIs become more integrated into medical practice, the privacy of brain data and informed consent must be handled with great care. The sensitive nature of neural information demands clear and transparent regulations to prevent misuse. It is also crucial to ensure equitable access to these technologies, so that people from all socioeconomic backgrounds can benefit from advancements in neurotechnology. Transparency in the use of AI algorithms and accountability in data management will be essential to maintaining the trust of patients and users of these systems [93,94]. To address these concerns, experts recommend establishing globally coordinated ethical guidelines, implementing new privacy measures including “Neurorights”, developing methods to prevent bias, and adopting guidelines for equitable distribution of neurotechnological devices [95].

The analysis presented in this manuscript is highly relevant to the scientific community interested in EEG-BCI, as it not only provides a comprehensive overview of current trends but also lays a strong foundation for future research. By identifying key players and emerging research areas, it helps researchers and decision-makers gain a clearer understanding of the field and direct their efforts toward areas with the greatest potential impact. This study also highlights how advancements in technology and interdisciplinary collaborations are transforming the use of EEG-BCI in clinical rehabilitation, positioning it as a key reference for future developments, particularly in the context of personalized treatments through artificial intelligence and big data. The visibility of BCIs relies heavily on the dissemination of research, and conducting updated bibliometric analyses is crucial to the continuous advancement of BCIs and their role in improving patient outcomes.

This work, compared to the bibliometric analysis by Li et al., 2023 [47], presents the following strengths: it provides broader coverage of rehabilitation areas (motor, sensory, speech, and cognitive), utilizes multiple databases for more comprehensive data collection, and emphasizes the role of AI in enhancing BCI systems. In contrast, the Li et al. [47] analysis focuses mainly on post-stroke rehabilitation and relies solely on the WoS Core Collection, which limits its scope and perspective in the broader field of BCI.

Finally, this study has some limitations: the bibliometric analysis was based on specific databases, which may have excluded relevant publications not indexed in these sources. Additionally, the focus on English-language publications may have restricted the inclusion of significant research in other languages. Future research should expand the scope to include a wider variety of databases and consider a broader linguistic diversity to obtain a more comprehensive view of the state of BCI research.

## 5. Conclusions

This comprehensive bibliometric analysis of EEG-based BCIs in rehabilitation provides valuable insights into the field’s growth, key contributors, and emerging trends over the past decade. The study highlights a significant increase in scientific output since 2016, particularly in motor and sensory rehabilitation, with a strong focus on utilizing EEG-BCIs for clinical applications. The growing prominence of terms related to neuroplasticity, motor imagery, and gender differences reflects a shift toward more personalized and sophisticated methodologies in neurorehabilitation. Global contributions to EEG-BCI research are led by countries such as China, the United States, and Germany, indicating strong international collaboration, particularly in engineering, neuroscience, and clinical fields. However, regions like Latin America remain underrepresented, underscoring the need for increased investment, infrastructure development, and collaboration in neurotechnology research. The study also emphasizes the potential of artificial intelligence and machine learning to revolutionize BCI systems, making them more adaptive and tailored to individual needs, while big data analytics can provide long-term insights into patient outcomes. Despite these technological advancements, ethical considerations, such as data privacy and equitable access to these innovations, remain crucial to ensure responsible development and application.

## Figures and Tables

**Figure 1 sensors-24-07125-f001:**
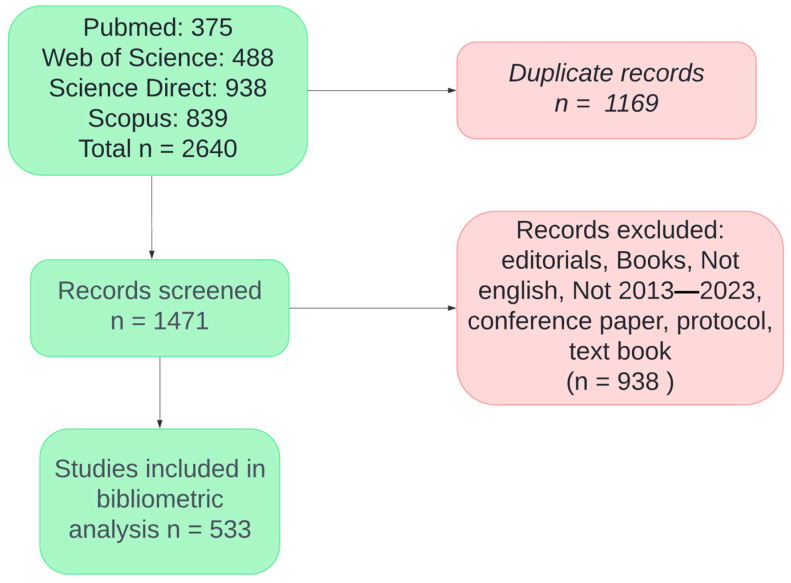
Flowchart of Article Selection for Bibliometric Analysis. The detailed process of selection and enrollment involved two authors manually reviewing the abstracts and full texts of the articles. Articles deemed irrelevant to the topic were excluded.

**Figure 2 sensors-24-07125-f002:**
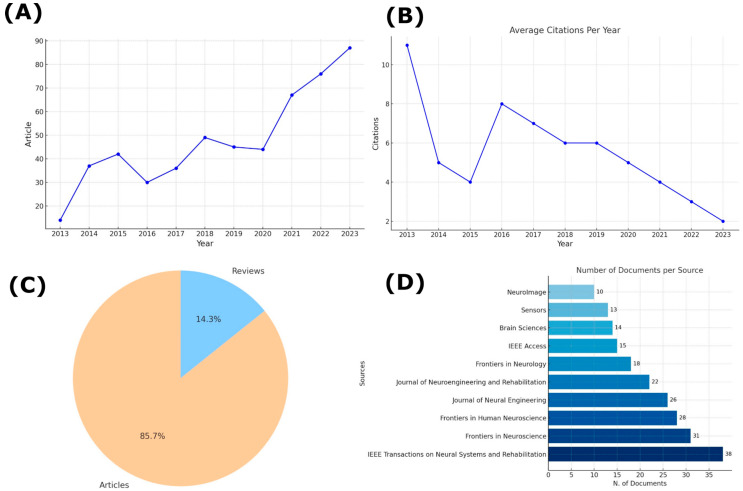
Document Analysis Over the Period from 2013 to 2023. (**A**) Number of articles published per year. (**B**) Average citations per year. (**C**) Distribution of document types between articles and reviews. (**D**) Number of documents per source, indicating the top 10 publishing journals.

**Figure 3 sensors-24-07125-f003:**
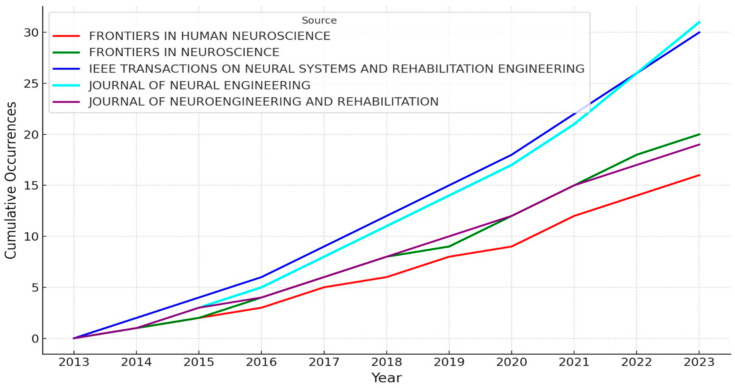
Cumulative Occurrences by Source Over Time. Cumulative number of publications per top five journal sources from 2013 to 2023.

**Figure 4 sensors-24-07125-f004:**
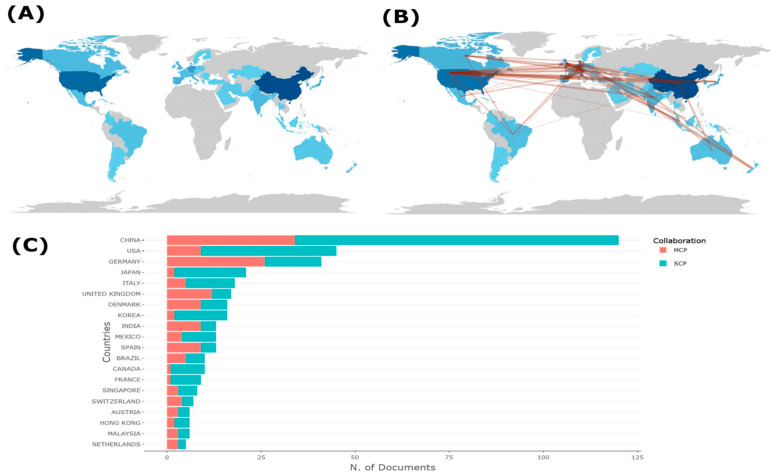
Scientific Output and Collaboration by Country. (**A**) Global distribution of scientific production, with darker shades representing countries with higher output. (**B**) International collaboration networks, visualized by the connections between countries through co-authorship and joint research projects, with the brown line specifically indicating the pathways and interactions of these collaborations across different regions. (**C**) Number of documents by country, differentiated by Single-Country Publications (SCP) and Multiple-Country Publications (MCP), showing the collaboration type for each country.

**Figure 5 sensors-24-07125-f005:**
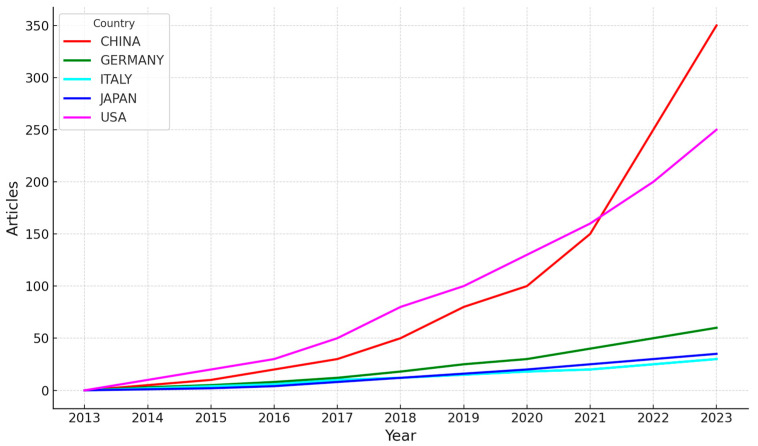
Growth of Scientific Publications by Top Five Countries (2013–2023).

**Figure 6 sensors-24-07125-f006:**
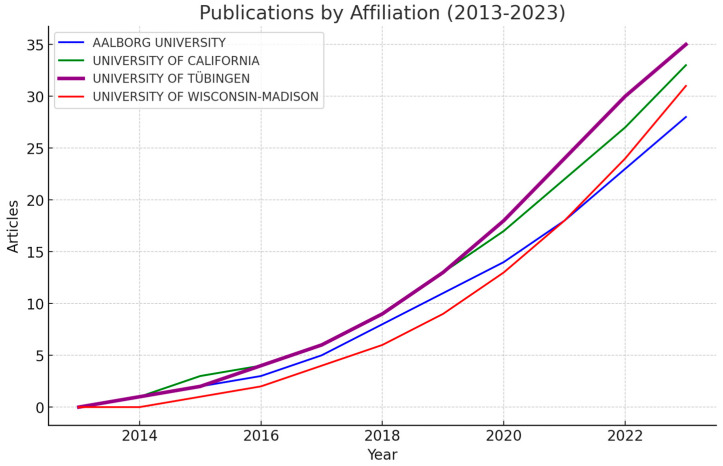
Article Publication Trends by the Top Four Affiliations (2013–2023).

**Figure 7 sensors-24-07125-f007:**
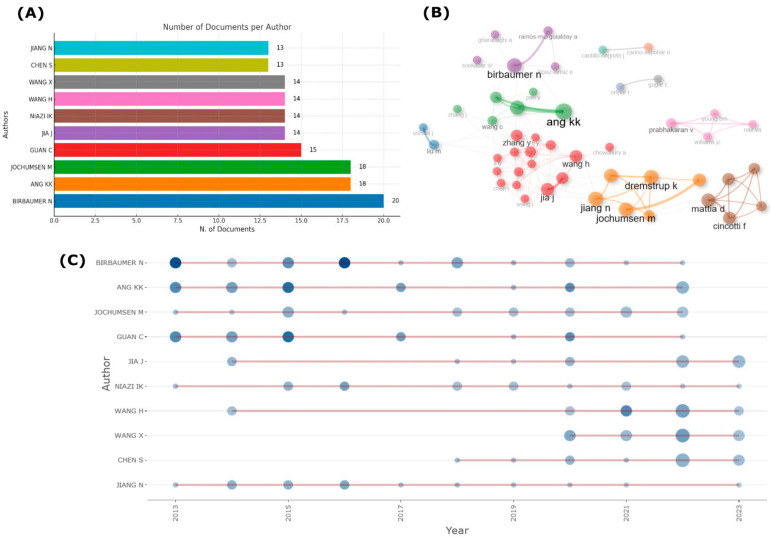
Scientific Output and Collaboration by Author. (**A**) Number of documents published per author, showing individual contributions to the field. (**B**) Author collaboration network, illustrating connections between authors through co-authorship and collaborative research. (**C**) Author productivity over time (2013–2023), highlighting the number of publications each author has contributed throughout the years.

**Figure 8 sensors-24-07125-f008:**
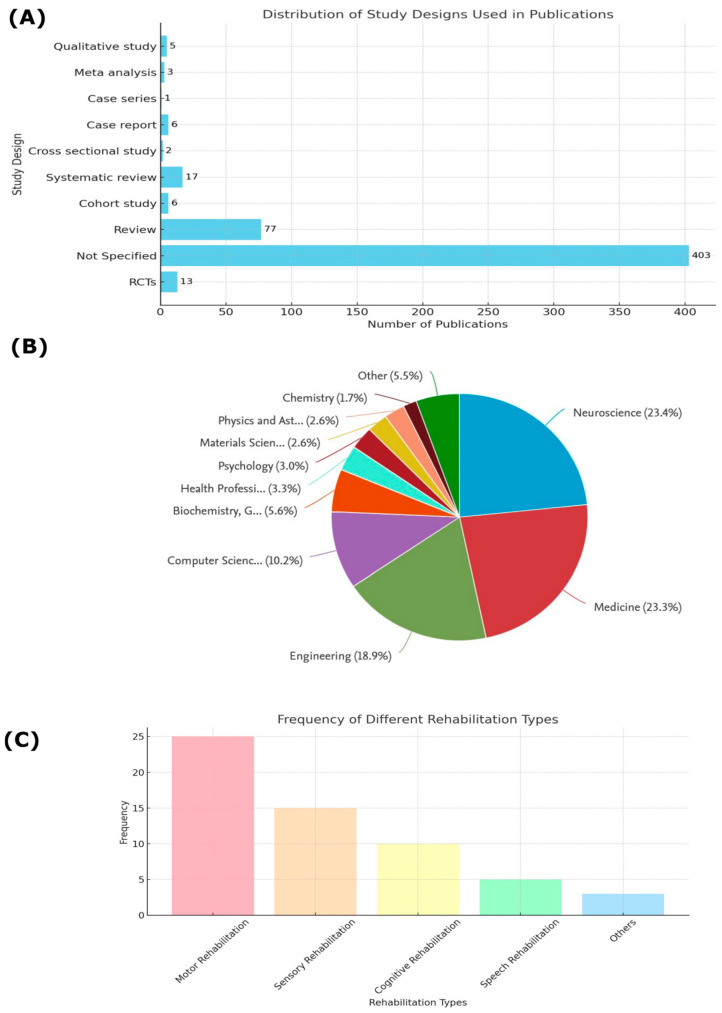
Study Designs, Fields of Research, and Rehabilitation Types. (**A**) Distribution of study designs used in the analyzed publications, showing the frequency of different study methodologies. (**B**) Proportional distribution of the academic fields contributing to BCI research, including neuroscience, medicine, and engineering, among others. (**C**) Frequency distribution of different rehabilitation types addressed in the research, with motor rehabilitation being the most common focus. RCTs; Randomized Clinical Trials.

**Figure 9 sensors-24-07125-f009:**
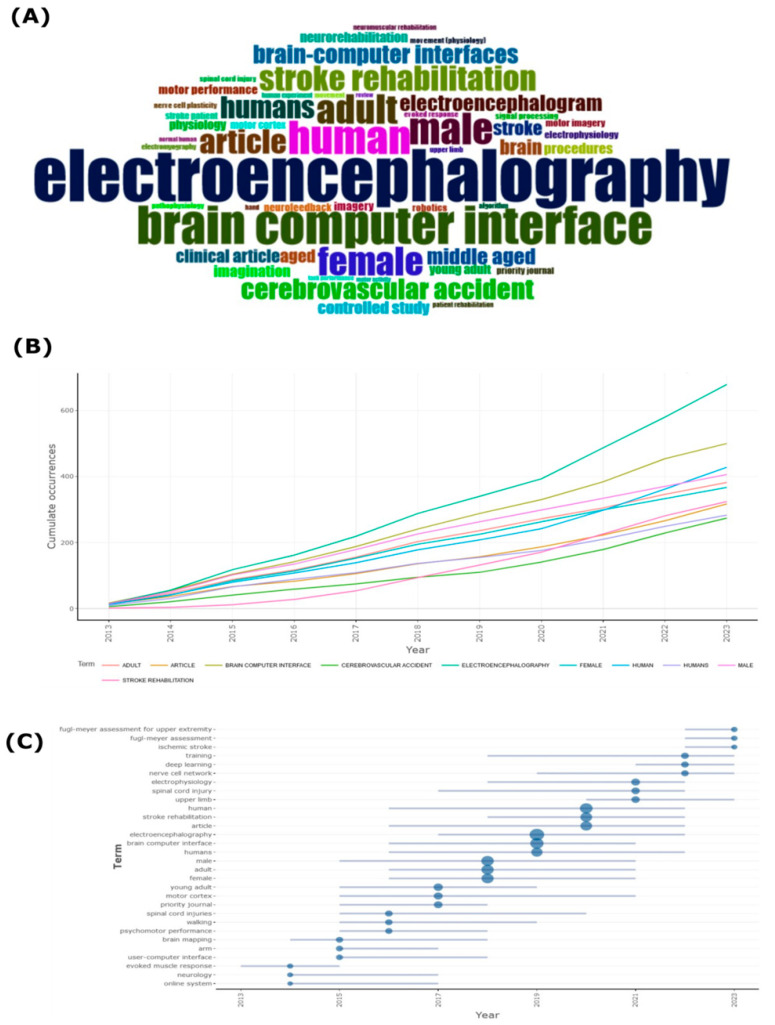
Keyword Analysis and Trends in Publications. (**A**) Word cloud of the most frequent keywords in publications. (**B**) Cumulative occurrences of selected keywords over the period from 2014 to 2024. (**C**) Timeline of emerging keywords in publications.

**Figure 10 sensors-24-07125-f010:**
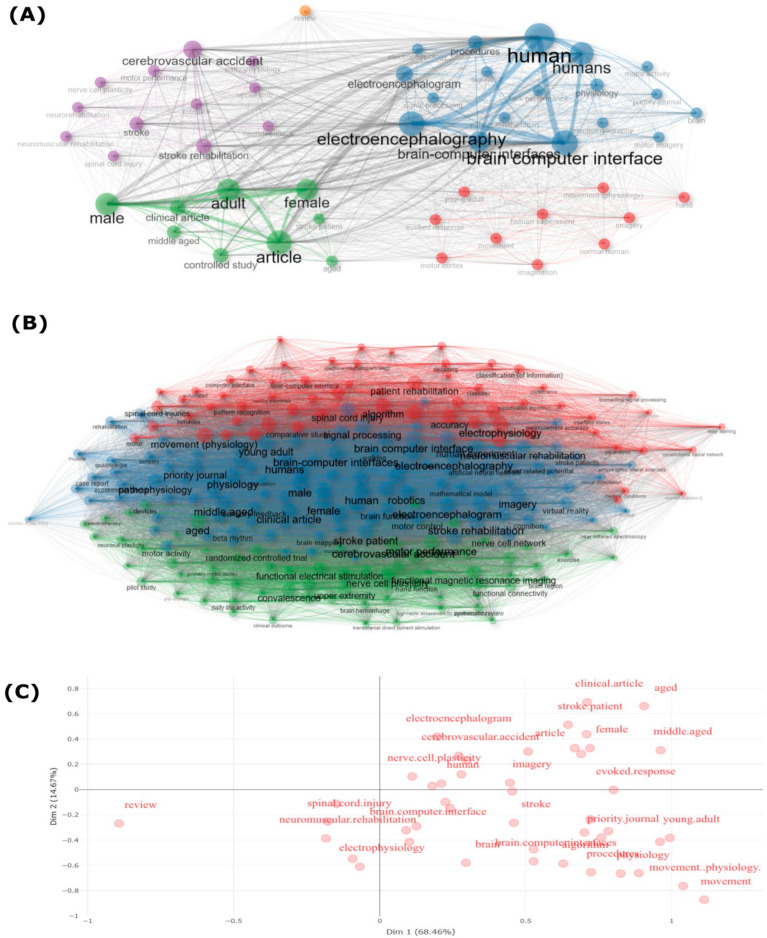
Thematic and Factorial Analysis of BCI Research Keywords. (**A**) Thematic map illustrating the development and relevance of research themes in BCI. (**B**) Factorial analysis plot showing keyword relationships and clustering in BCI research. (**C**) Dimensional analysis of keyword distributions, mapping terms across two dimensions to highlight their contextual relevance and co-occurrence patterns within BCI studies.

**Table 2 sensors-24-07125-t002:** Institutions by Number of Published Articles.

Top 25 Affiliation	Articles
University of Wisconsin–Madison	56
Aalborg University	30
University of Tübingen	30
University of California	27
Fudan University	21
Keio University	15
Sapienza University of Rome	15
Tianjin University	15
Institute for Infocomm Research	14
Pirogov Russian National Research Medical University	14
Scms School of Engineering and Technology	14
Tsinghua University	14
Eberhard Karls University Tuebingen	12
New Zealand College of Chiropractic	12
Keio university School of Medicine	11
Schiedlberg	11
Shanghai Jiao Tong University	11
Institute of Higher Nervous Activity and Neurophysiology	10
Nanyang Technological University	10
University of Calgary	10
University of Essex	10
Beihang University	9
University of Waterloo	9
Xi’an Jiaotong University	9

## Data Availability

Data are contained within the article.

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
