# Peer review of "Electroencephalography-Based Brain-Computer Interfaces in Rehabilitation: A Bibliometric Analysis (2013–2023)"

_sensors, 2024, doi:10.3390/s24227125_

Round 1

Reviewer 1 Report

Comments and Suggestions for Authors

I like the intent of the manuscript but there are several flaws those need to be corrected.

34- BCIs - full form

34 - such as stroke, amyotrophic lateral sclerosis (ALS), and spinal cord injuries [1,2] - I recommend citing a paper for each disorder next to it.

39-  these diseases - change to disorders

85 - Given that your search has Electroencephalography specifically maybe you focus the review on EEG based BCIs, rather talking about BCIs in general/. Else I would recomend using EEG+BCI and MEG+BCI etc. to have a more comprehensive search.

148- slight decline - seems significant decline. Although since you are focusing on EEG it makes sense. There have been increased use of other non-invasive modalities like MEG and several invasive research. So it might be true for EEG based BCI research but doesn't show the trend of all BCI research. Thus, In would recommend again to do a focused review on EEG or enhance the parameter search.

VIP: I suggest to remove 2024 from your graph because that's the data for 4 months only. Either interpolate the possible number of publications then fit or only show up to 2023;.

- Figure 3B seems wrong. you have about 800 papers but the pargraph shows about 200 papers total. are you showing for a particular year?

- Figure 4- please use a colorbar to represent the numbersor %

- Criteria for key publication- first author h-index and citation is not appropriate. If an article is published earlier it is likely to have higher citations compared to an impactful paper published in 2024 April. Similarly, first author h-index is not appropriate to judge the impact of a paper.  I would remove table 1 or use top5 or 10 cited papers, rather than describing them as key papers. Probably better to show a bar graph of journals and a number of publications and citations.

- Figure 6 - using top 10 affiliation doesn't show the clear picture. For the bar graph it's alright, or else you will have too many affiliations, but for the correlation between h-0index and citations I would use all data except "recent" papers.

-Figure 7 should be merged with Table 1 like I pointed out earlier

- Keyword analysis is flawed. Every human subject research paper will have demographics such as human, male , or female in their manuscript which doesn't signify that these words have an impact in BCI research. Further, since you used electroencephalography in your search criteria that would show up most often. I will remove this part or use the abstract only to show a word map.

- 329 - Electroencephalography (EEG) is the most frequently used method, significantly outnumbering other BCI techniques.- Same issue as before. either do a comprehensive search without using EEG in your search criteria or focus on review on EEG and this part is obsolete. 

- Same issue for table 2 with EEG

- Analysis of Leading Authors - It should be done with full name basis. There is a high likelihood that Wang J might correspond to abbreviation of different authors such as Wang Jun or Wang Jue or Wang Jiyao

- Figure 11- Please see my comment for keyword analysis.

- Figure 12 A - Issue of EEG persists

- The purpose of this study is to provide a comprehensive overview of the current state of research on BCIs in the field of rehabilitation over the past ten years, using bibliometric tools. - Not true. focus your review with EEG only or increase the search criteria

Next the entire discussion and conclusion is based on flawed parameter search. I would suggest the authors to correct their seara\ch criteria and redo their analysis or do a focused review of EEG-BCIs for rehabilitation and amend the manuscript accordingly.

Author Response

We would like to express our sincere gratitude for your valuable comments and suggestions, which have undoubtedly contributed to the improvement of our article. We have carefully considered each of your points and have made the necessary changes, which are highlighted in yellow for easy reference.

Additionally, following the suggestion of a previous reviewer, we have completely reoriented the focus of the article to center on EEG-based Brain-Computer Interfaces (BCIs), as this better emphasizes the impact of our review on neurocognitive rehabilitation. This shift in focus reflects the growing relevance of this technology in current research.

I like the intent of the manuscript but there are several flaws those need to be corrected.

34- BCIs - full form

Response: It was corrected

34 - such as stroke, amyotrophic lateral sclerosis (ALS), and spinal cord injuries [1,2] - I recommend citing a paper for each disorder next to it.

Response: It was corrected

39-  these diseases - change to disorders

Response: It was corrected

85 - Given that your search has Electroencephalography specifically maybe you focus the review on EEG based BCIs, rather talking about BCIs in general/. Else I would recomend using EEG+BCI and MEG+BCI etc. to have a more comprehensive search.

Response: The article was rewritten according to your recommendations.

148- slight decline - seems significant decline. Although since you are focusing on EEG it makes sense. There have been increased use of other non-invasive modalities like MEG and several invasive research. So it might be true for EEG based BCI research but doesn't show the trend of all BCI research. Thus, In would recommend again to do a focused review on EEG or enhance the parameter search.

Response: The article was rewritten according to your recommendations.

VIP: I suggest to remove 2024 from your graph because that's the data for 4 months only. Either interpolate the possible number of publications then fit or only show up to 2023;.

Response: The article was rewritten according to your recommendations from 2013 to 2023.

- Figure 3B seems wrong. you have about 800 papers but the pargraph shows about 200 papers total. are you showing for a particular year?

Response: The figure 3 was removed as the search parameters were changed and the article was rewritten, leaving only figure 4 for the geographic analysis.

- Figure 4- please use a colorbar to represent the numbersor %

Response: The bar you mentioned cannot be added as Bibliometrix does not have it predefined in the graph.

- Criteria for key publication- first author h-index and citation is not appropriate. If an article is published earlier it is likely to have higher citations compared to an impactful paper published in 2024 April. Similarly, first author h-index is not appropriate to judge the impact of a paper.  I would remove table 1 or use top5 or 10 cited papers, rather than describing them as key papers. Probably better to show a bar graph of journals and a number of publications and citations.

Response: The table has been adjusted by removing the H-index and applying the other suggested modifications. The revised Table 1 now includes the following columns: Authors and Year of publication, Manuscript, Country, Affiliation, Funding, Type of study, Journal, Quartile, and Citations. Additionally, the graph showing the number of publications per journal is presented in Figure 2.

- Figure 6 - using top 10 affiliation doesn't show the clear picture. For the bar graph it's alright, or else you will have too many affiliations, but for the correlation between h-0index and citations I would use all data except "recent" papers.

Response: For the bar graph, we will maintain the top 10 affiliations to avoid overcrowding the visualization. Additionally, the H-index has been removed in accordance with your suggestions.

-Figure 7 should be merged with Table 1 like I pointed out earlier

Response: Due to the distinct nature of the topics, it is not possible to merge Figure 7 with Table 1. Additionally, Figure 3 now shows the top 5 cumulative occurrences by source over time, specifically the cumulative number of publications per top 5 journal sources from 2013 to 2023.

- Keyword analysis is flawed. Every human subject research paper will have demographics such as human, male , or female in their manuscript which doesn't signify that these words have an impact in BCI research. Further, since you used electroencephalography in your search criteria that would show up most often. I will remove this part or use the abstract only to show a word map.

Response: It's important to clarify that this current study focuses specifically on EEG-based BCI research, which distinguishes it from broader BCI topics. In this context, terms like "human," "male," and "female" are indeed relevant, as they influence the design and applicability of BCI systems, particularly in areas like neurorehabilitation and personalized medicine. Understanding demographic factors is crucial for tailoring BCI technologies to different user populations, which can directly impact system performance and user experience.

Additionally, even though "electroencephalography" was part of the search criteria, it remains an essential term in EEG-based BCI research. The frequency of this term in keyword analysis reflects its central role in this specific BCI approach. For these reasons, we believe that the inclusion of such keywords offers meaningful insights into the scope and trends of the field.

- 329 - Electroencephalography (EEG) is the most frequently used method, significantly outnumbering other BCI techniques.- Same issue as before. either do a comprehensive search without using EEG in your search criteria or focus on review on EEG and this part is obsolete. 

Response: The manuscript has been entirely rewritten.

- Same issue for table 2 with EEG

Response: The manuscript has been entirely rewritten.

- Analysis of Leading Authors - It should be done with full name basis. There is a high likelihood that Wang J might correspond to abbreviation of different authors such as Wang Jun or Wang Jue or Wang Jiyao

Response: Response: The manuscript has been entirely rewritten.

- Figure 11- Please see my comment for keyword analysis.

Response: Response: The manuscript has been entirely rewritten.

- Figure 12 A - Issue of EEG persists

Response: Response: The manuscript has been entirely rewritten.

- The purpose of this study is to provide a comprehensive overview of the current state of research on BCIs in the field of rehabilitation over the past ten years, using bibliometric tools. - Not true. focus your review with EEG only or increase the search criteria

Response: Response: The manuscript has been entirely rewritten.

Next the entire discussion and conclusion is based on flawed parameter search. I would suggest the authors to correct their seara\ch criteria and redo their analysis or do a focused review of EEG-BCIs for rehabilitation and amend the manuscript accordingly.

Response: The manuscript has been entirely rewritten.

Reviewer 2 Report

Comments and Suggestions for Authors

1. It is suggested to align the small labels “(A)” and “(B)” in the figures.

2. The summary of the article is not comprehensive enough. It is recommended that the authors highlight important findings and include further reflections on this work.

3. The article reviews research from the past decade. It is suggested to use more text to elaborate on recommendations and insights for future development.

4. The significance of the research in the article is limited. It is recommended to provide a more powerful explanation to demonstrate the research significance.

5. The article contains a lot of content. It is suggested to create a chart to clearly describe the research content of each section before detailing the experiments.

6. Figure 11 is somewhat blurry. It is recommended to use a clearer image or consider alternative presentation methods.

Comments on the Quality of English Language

The grammar and typos needed to be improved for understanding.

Author Response

We would like to express our sincere gratitude for your valuable comments and suggestions, which have undoubtedly contributed to the improvement of our article. We have carefully considered each of your points and have made the necessary changes, which are highlighted in yellow for easy reference.

Additionally, following the suggestion of a previous reviewer, we have completely reoriented the focus of the article to center on EEG-based Brain-Computer Interfaces (BCIs), as this better emphasizes the impact of our review on neurocognitive rehabilitation. This shift in focus reflects the growing relevance of this technology in current research.

  1. It is suggested to align the small labels “(A)” and “(B)” in the figures.

Response: It was corrected

  1. The summary of the article is not comprehensive enough. It is recommended that the authors highlight important findings and include further reflections on this work.

Response: It was corrected

  1. The article reviews research from the past decade. It is suggested to use more text to elaborate on recommendations and insights for future development.

Response: It was corrected

  1. The significance of the research in the article is limited. It is recommended to provide a more powerful explanation to demonstrate the research significance.

Response: The manuscript has been entirely rewritten according your suggestions.

  1. The article contains a lot of content. It is suggested to create a chart to clearly describe the research content of each section before detailing the experiments.

Response: It was clarified according to your suggestions.

  1. Figure 11 is somewhat blurry. It is recommended to use a clearer image or consider alternative presentation methods.

Response: The image has been improved, and its significance has been further elaborated.

Round 2

Reviewer 1 Report

Comments and Suggestions for Authors

Thank you to the authors for revising your manuscript as per my recommendations. I have 2 minor concerns which need to be addressed.

1. Figure 2B doesn't seem right. Are you sure about the y-axis for number of citations? Based on your figure there are only 2 citations in the field in 2023? That couldn't be the case. Please recheck.

2. Some of your figures are screenshots and blurry. Please use high quality images. 

Author Response

  1. Figure 2B doesn't seem right. Are you sure about the y-axis for number of citations? Based on your figure there are only 2 citations in the field in 2023? That couldn't be the case. Please recheck. Response: 

    Thank you for your feedback regarding Figure 2B. Upon careful examination, the low citation count for 2023 can be attributed to several well-established factors in bibliometric analysis. Firstly, there is an inherent time lag between the publication of an article and its subsequent citation by other researchers. Articles published in 2023 have not yet had sufficient time to be widely cited, which is a common phenomenon observed in bibliometric studies across various disciplines.

    Additionally, the y-axis of Figure 2B represents the average number of citations per article for each publication year. This metric naturally yields lower values for more recent years, as those articles have not accumulated citations over time. The increasing number of publications in recent years, as depicted in Figure 2A, also plays a role. A higher volume of recent articles can initially lead to a lower average citation count per article until these works gain recognition and are cited in future research.

2. Some of your figures are screenshots and blurry. Please use high quality images.  Response: They were updated in image quality.
